# Targeted Therapy Development in Acute Myeloid Leukemia

**DOI:** 10.3390/biomedicines11020641

**Published:** 2023-02-20

**Authors:** Tulasigeri M. Totiger, Anirban Ghoshal, Jenna Zabroski, Anya Sondhi, Saanvi Bucha, Jacob Jahn, Yangbo Feng, Justin Taylor

**Affiliations:** 1Sylvester Comprehensive Cancer Center, University of Miami Miller School of Medicine, Miami, FL 33136, USA; 2Department of Molecular and Cellular Pharmacology, University of Miami Miller School of Medicine, Miami, FL 33136, USA

**Keywords:** targeted therapy, myeloid neoplasm, precision medicine, acute myeloid leukemia, drug development, small molecule inhibitor, selective drugs, non-selective drugs

## Abstract

Therapeutic developments targeting acute myeloid leukemia (AML) have been in the pipeline for five decades and have recently resulted in the approval of multiple targeted therapies. However, there remains an unmet need for molecular treatments that can deliver long-term remissions and cure for this heterogeneous disease. Previously, a wide range of small molecule drugs were developed to target sub-types of AML, mainly in the relapsed and refractory setting; however, drug resistance has derailed the long-term efficacy of these as monotherapies. Recently, the small molecule venetoclax was introduced in combination with azacitidine, which has improved the response rates and the overall survival in older adults with AML compared to those of chemotherapy. However, this regimen is still limited by cytotoxicity and is not curative. Therefore, there is high demand for therapies that target specific abnormalities in AML while sparing normal cells and eliminating leukemia-initiating cells. Despite this, the urgent need to develop these therapies has been hampered by the complexities of this heterogeneous disease, spurring the development of innovative therapies that target different mechanisms of leukemogenesis. This review comprehensively addresses the development of novel targeted therapies and the translational perspective for acute myeloid leukemia, including the development of selective and non-selective drugs.

## 1. Introduction

Acute myeloid leukemia (AML) is a hematologic malignancy that is characterized by excessive growth of immature white blood cells (myeloblasts) in the bone marrow and in circulation, intrinsically affecting the process of hematopoiesis [1]. AML is a complex and heterogeneous neoplasm that progresses rapidly and can be fatal within weeks to months, if not treated [2,3,4,5]. Patients with this disease show fatigue, shortness of breath, easy bruising, and bleeding [6]. They have a greater risk of infection, and myeloblasts may disseminate to the gums, skin, and brain [6,7]. According to recent data from the National Cancer Institute (NCI), the rate of occurrence of new cases of acute myeloid leukemia (AML) per year was 4.1 per 100,000 men and women, with a death rate of 2.7 per 100,000 men and women [8].

AML has several sub-types defined by cytogenetic and molecular features, making the disease very complex to treat and requiring that therapeutics possess multi-targeting functionality [9,10,11,12,13]. The genomic landscape of AML has revealed a wide range of mutations falling into different functional categories of genes that play pivotal roles in AML pathogenesis [13,14]. The most common mutation of AML is in *FLT3,* which is attributed to the signaling gene category, along with mutations in *KIT, PTPN11, RAS, JAK2,* and *PPM1D* [15,16]. The splicing factor gene group contains *SF3B1, SRSF2, U2AF1,* and *ZRSR2* mutations [17,18], while the *NPM1* gene mutation group contains only mutation *NPM1* [10,19]. *TP53, WT1,* and *PHF6* mutations form the tumor suppressor gene category [20,21]. Additionally, the genes of DNA methylation exhibited *DNMT3A, TET2, IDH1,* and *IDH2* mutations. Chromatin modifier genes showed *ASXL1, EZH2,* and *KMT2A,* and lastly, cohesin complex genes contained *STAG1, STAG2, RAD21, SMC1A,* and *SMC3* mutations [22].

## 2. Standard Treatment of AML and Resistance Mechanisms

The standard frontline treatment for AML consists of two phases [23]. The first phase, called remission induction, involves treatment with two chemotherapies—cytarabine, and either daunorubicin or idarubicin. In certain situations, a third drug may be added in this phase to achieve better results for patients with particular genetic alterations. For patients with an *FLT3* mutation, midostaurin, a multikinase agent with activity against FLT3, should be added. Gemtuzumab ozogamicin, an antibody-drug conjugate targeting the CD33 protein, is added for patients with favorable risk cytogenetics (inversion 16, translocation 8;21, or translocation 16;16). Although uncommon, if the disease has crossed the blood–brain barrier into the central nervous system, additional chemotherapy is delivered into the cerebrospinal fluid (CSF) [10,24]. The second phase of treatment is called consolidation, or post-remission therapy, which targets and destroys any remaining cells to prevent relapse. Consolidation therapy for patients who are fit enough to receive high-dose chemotherapy involves several cycles of higher doses of cytarabine, followed by an allogeneic stem cell transplant for patients with an increased risk of relapse. Consolidation therapy for less fit patients with underlying health conditions consists of a lower dosage of cytarabine than healthy patients, as well as the consideration of a non-myeloablative stem cell transplant [25,26,27].

Despite the best treatment with standard therapies, more than 60% of AML patients across all age categories will eventually relapse [28]. Additionally, standard chemotherapies can have many adverse effects in both the short and long-term, especially in older individuals. Previously, older AML patients have had limited treatment options, relying purely on hypomethylating agents [26,29]. Recently, however, the addition of venetoclax, a BCL2 inhibitor, to hypomethylating agents was shown to be superior to hypomethylating agents alone [30]. Other treatment options that are currently being studied are chimeric antigen receptor (CAR) T cell therapy [31,32], immunotherapy comprising inhibition of immune negative regulators, and antibody-based treatment, which have the potential to revolutionize the treatment landscape to address the complex and heterogenous aspects of AML [33,34]. This review will focus on the past and future development of targeted small-molecule therapies.

## 3. History of Small Molecule Development in Acute Myeloid Leukemia

The classical anthracycline and cytarabine-based chemotherapy for AML lacks durable efficacy and has poor tolerability in older patients [35]. This unmet medical need demands the development of novel therapeutics for AML. Based on prognostic studies of AML derived from clinical outcomes of molecularly defined subsets and from mechanistic disease studies in laboratory models, promising targets for molecularly targeted therapy have been identified in this disease. This marked the beginning of the development of small molecule drugs through clinical trials using these predictive genetic markers [36].

In the last decade, a better understanding of the heterogeneity and pathophysiology of the disease has led to promising new therapies, resulting in numerous U.S. Food and Drug Administration (FDA)-approved drugs [37]. The FDA-approved drugs for AML, beyond standard chemotherapy, include gemtuzumab ozogamicin [38], hypomethylating agents [39], FLT3 (Fms-like tyrosine kinase 3) inhibitors [40], IDH (isocitrate dehydrogenase) inhibitors [41], venetoclax [42], CPX-351 (liposomal cytarabine and daunorubicin), also called Vyxeos [43], and hedgehog pathway inhibitors [44]. We will discuss these drugs further in the following sections.

### 3.1. Approved Drugs

Since the 1970s, the classical therapy for AML has consisted of cytarabine combined with an anthracycline (daunorubicin or idarubicin), famously known as the “7 + 3” regimen (Figure 1) [45]. Over the last decade, the FDA has approved numerous new drugs (Figure 1) [37]. The small-molecule FDA-approved drugs for AML, beyond standard chemotherapy, include IDH inhibitors [41] (olutasidenib, ivosidenib, enasidenib), FLT3 inhibitors [40] (gilteritinib, midostaurin), BCL-2 inhibitor [42] (venetoclax), hypomethylating agents [39] (azacitidine, decitabine), and CPX-351 (liposomal cytarabine and daunorubicin) [43].

#### 3.1.1. IDH Inhibitors

Isocitrate dehydrogenase (IDH) inhibitors have emerged as a promising therapeutic strategy for the treatment of acute myeloid leukemia. The enzymes IDH1 and IDH2 are responsible for the conversion of isocitrate to alpha-ketoglutarate within the TCA cycle [46]. In 2018, the FDA approved ivosidenib for relapsed/refractory *IDH1*-mutated AML [47]. This drug is used in monotherapy with the dose of 500 mg PO qDay, until disease progression or unacceptable toxicity. The small molecule IDH2 inhibitor enasidenib, active against *IDH2* R140 and *IDH2* R172 variants [41], was approved by the FDA in 2017 for relapsed/refractory IDH2-mutated AML [48], with the advantage of its efficacy in certain types of acute myeloid leukemia (AML) against reduced white blood cells or relapse, even after the treatment with previously approved chemotherapy medications. The optimal use of this drug is 100 milligrams (mg) once a day. In December of 2022, another drug, olutasidenib, was approved for adult patients with relapsed or refractory AML with a susceptible *IDH1* mutation, as detected by an FDA-approved test [49]. This drug was approved for the treatment of patients with treatment-naïve and relapsed or refractory acute myeloid leukemia (AML). It was approved on the basis of positive results from the 2102-HEM-101 (NCT02719574) study. The optimal dosage of this drug is 150 mg, twice daily. Based on their better efficacies with the consideration of less toxicity, *IDH* inhibitors are being used for specific patients, based on their biological phenotype regarding white blood cells and the patients’ response to the previously approved drugs, with a strong commitment to seek the best outcomes for patients of AML. IDH inhibitors can block mutant IDH proteins to help leukemia cells differentiate normally, and for this reason, they can also be called differentiation agents. In a study of ivosidenib with subcutaneous or intravenous chemo (azacitidine), common side effects include febrile neutropenia (28% ivosidenib vs. 34% placebo) and neutropenia (27% ivosidenib vs. 16% placebo) [50]. IDH inhibitors in general are well-tolerated in AML patients, as they induce differentiation of AML cells, which leads to the on-target side effect of differentiation syndrome in up to 20% of patients [51].

#### 3.1.2. FLT3 Inhibitors

The most frequent mutations in AML are those in the gene for FMS-like tyrosine kinase 3 (*FLT3)*, which is found in one-third of patients with de novo AML [52]. Over the last twenty years, several different FLT3 inhibitors have been developed, classified into first and second-generation inhibitors. These drugs were approved to act against the ligand that promotes the proliferation and survival of leukemic blasts that express the FLT3 receptor. These drugs are kinase-specific and have shown promising effects in the treatment of AML patients with *FLT3* mutations. Midostaurin and sorafenib belong to the class of first-generation FLT3 inhibitors, whereas quizartinib and gilteritinib belong to the class of second-generation inhibitors. The second-generation inhibitors were developed with a potential need to avoid off-target effects, such as nausea, febrile neutropenia, mucositis, vomiting, headache, petechiae, and fever. Thus far, the second-generation inhibitors currently in trials have shown promising effects in the treatment of *FLT3*-mutated AML [53]. Two FLT3 inhibitors, midostaurin (first generation) and gilteritinib (second generation), have recently been approved by the U.S. Food and Drug Administration (FDA) for use in patients with FLT3-mutated AML [52]. Midostaurin was also approved for the treatment of adults with aggressive systemic mastocytosis (SM), SM with associated hematological neoplasm, or mast cell leukemia. The second-generation FLT3 inhibitor gilteritinib was approved for the treatment of adult patients who have relapsed or refractory acute myeloid leukemia (AML) with an *FLT3* mutation, as detected by an FDA-approved test. Overall, these FLT3 inhibitors represent a new standard of care for patients with FLT3 mutations in both the first-line and salvage settings.

#### 3.1.3. BCL2-Inhibitors

BCL-2 inhibitors are the class of drugs which selectively induce apoptosis in cancerous cells. These have been used extensively in hematologic malignancies alone or in combination with other chemotherapies. Previously, BH3-mimetics, such as ABT-737 and navitoclax (ABT-263), that comprise a novel class of BCL-2 inhibitors have shown promising results in several hematological malignancies, both alone and in combination with other cancer therapies. However, these inhibitors exhibited some inadequacies, such as insufficient bioavailability and on-target toxicity of BCLxL inhibition. Hence, the other BCL-2 inhibitors came into existence to fill this gap [54,55]. BCL-2 has been indicated to be a critical regulatory protein that monitors apoptotic and cell cycle regulation events. When BCL-2 mutations are present, BCL-2 can be greatly overexpressed, leading to chemotherapeutic resistance and a poorer prognosis. These inhibitors have the capacity to circumvent significant thrombocytopenia due to concomitant inhibition of BCL-XL, thus making them better clinical entities for the treatment of AML [56,57]. Venetoclax is a potent and orally active small molecule inhibitor targeting B-cell lymphoma 2 (BCL-2) [58]. This plays an important role in intrinsic apoptosis. Mechanistically, this drug acts through the induction of pro-apoptotic cell death. The successful treatment with this drug against chronic lymphocytic leukemia (CLL) led to its first FDA approval in 2016 for patients with 17p-deleted CLL, and it was later granted accelerated approval for use in acute myeloid leukemia in 2018, with full approval in 2021. Venetoclax monotherapy was successfully tolerated in a phase 2 trial of 32 relapsed/refractory AML patients, leading to a 19% combined complete response with incomplete count recovery rate [59]. The addition of a hypomethylating agent greatly increased the response rates in combination with venetoclax and is discussed later in this review. Adverse effects included cytopenia and febrile neutropenia in the early days, and the (30-day) mortality rate was 3%, as revealed by one of the recent studies [60].

#### 3.1.4. Hypomethylating Agents

Hypomethylating agents (HMAs) are an important treatment option in patients with acute myeloid leukemia (AML) who are not suitable for intensive chemotherapy. To date, hypomethylating agents have been extensively tested in hematologic malignancies, due to their critical role in reversing harmful DNA aberrations in crucial regions. Recently, azacitidine was tested in two separate phase 3 studies in myelodysplastic syndrome (MDS) patients. Azacytidine is used either alone or in combination with other drugs for AML patients. This drug reduces the formation of leukemia cells and subsequently induces the bone marrow to produce more healthy cells. Mechanistically, azacytidine inhibits DNA methyltransferase by the formation of covalent bonds between DNA-cytosine methyltransferase and DNA containing 5-azacytosine, thus resulting in DNA hypomethylation [61]. Azacytidine was approved based on a randomized, multicenter, double-blind, placebo-controlled study (AG120-C-009, NCT03173248) that included 146 patients with newly-diagnosed AML with an IDH1 mutation who met at least one of the following criteria: age 75 years or older, baseline Eastern Cooperative Oncology Group performance status of 2, severe cardiac or pulmonary disease, hepatic impairment with bilirubin >1.5 times the upper limit of normal, creatinine clearance <45 mL/min, or other comorbidity. Decitabine is a DNA-hypomethylating agent that also induces differentiation and apoptosis of leukemic cells. It is a well-tolerated alternative to aggressive chemotherapy. Azanucleosides such as decitabine are chemically unstable, and precautions must be taken when identifying safe and effective methods to administer such drugs [62]. In general, decitabine has dual mechanisms of action. At low concentration, it promotes cellular differentiation. At high doses, it induces cytotoxic effects [63].

It has been found from the recent study that there is no significant difference between the azacytidine and decitabine arms in terms of the CR rate (17.5% vs. 19.2%, respectively; *p* = 0.78). Both groups also had similar median OS rates (8.7 months vs. 8.2 months; hazard ratio [4] for death = 0.97) (NCT02348489) [64]. This confirms that these drugs have nearly similar effects on AML patients. In a recent study involving 145 AML patients, azacytidine or decitabine, in combination with venetoclax, was effective and well tolerated in elderly patients with AML (NCT02203773) [60].

### 3.2. Non-Approved Drugs

Besides the FDA-approved drugs, some other FLT3 inhibitors have also undergone clinical trials for AML, namely sorafenib and quizartinib (Figure 2). Sorafenib is a multi-targeted tyrosine kinase inhibitor that mediates its anti-leukemic effect by inhibiting c-Kit activation of Erk in acute myeloid leukemia [65]. Quizartinib is a highly potent and specific FLT3 inhibitor with activity against *FLT3* wild-type and *FLT3-ITD*-mutated AML [66]. However, these drugs were not approved by the FDA, due to various concerns with the trial data [67]. Nevertheless, the need for more effective and better-tolerated therapies for AML demands attention. In this direction, research groups worldwide have proceeded to develop and refine the novel small molecules to combat acute myeloid leukemia. Among recent reports, in 2021, Wang et al. revealed an oxoindoline-based selective FLT3 inhibitor as a potential candidate drug against *FLT3-ITD*-positive AML, which is oftentimes correlated to an adverse prognosis [7,68]. According to the NCCN and ELN 2017 guidelines, *FLT3* mutations can evolve from diagnosis to relapse, suggesting testing for *FLT3-ITD* mutations at multiple time points throughout a patient’s disease course to help guide the most appropriate therapeutic decisions, prompting the resolution to address the unmet need for targeting *FLT3* mutations [69]. A pyrazolo-quinoline-based kinase inhibitor reported by Dayal et al. in 2021 was found to inhibit the proliferation of AML cells in vivo [70]. In 2022, Li et al. reported a first-in-class hydrazide-based HDAC inhibitor with an excellent pharmacokinetic profile and in vivo anti-AML activity (4 mg/kg p.o. TGI: 78.9%) [71]. In 2021, Han et al. reported a promising lead CDK9 inhibitor for the treatment of AML [72].

FLT3 is a transmembrane ligand-activated receptor tyrosine kinase (RTK) with a critical role in the primary stages of development for myeloid and lymphoid lineages. Additionally, the FLT3 ligand activates *FLT3* via various signaling pathways. FLT3 inhibitors have been slow to develop, as many are multi-targeted inhibitors are not selective for *FLT3* [73,74]. The first generation of FLT3 inhibitors were the nonselective repurposed tyrosine kinase inhibitors, lestaurtinib and midostaurin. However, these drugs had many significant off-target effects, such as VEGFR2 and c-KIT inhibition [53]. The second generation of sorafenib and crenolanib were also developed originally to target other receptors. Quizartinib (AC220) is a small molecule FLT3 tyrosine kinase inhibitor developed specifically as an FLT3 inhibitor. The administration of quizartinib resulted in the prolongation of the QT interval, which limited the dosage in patients. Other side effects included anemia, thrombocytopenia, and fatigue. This drug was otherwise well tolerated, with positive responses in 23 out of the 76 patients [73].

## 4. Novel Targeted Therapies in Development

The advent of new technological approaches in recent years has helped unravel the complexity of AML. Genomic approaches, including single-cell RNA sequencing, CRISPR gene editing, multi-targeted tissue imaging studies, and several other technological advances, have provided a better understanding of the complex genomic landscape of this disease. These recent developments have produced key discoveries in developing small molecular drugs that can potentially target specific genetic abnormalities or mutant proteins in AML (Figure 3). These targeted drugs can be used in conjunction with chemotherapy, or in cases when chemotherapy is not effective. In addition, novel targeted drugs were recently developed that have shown promising results against leukemic clone variations [75]. The identification of specific molecular targets for an individual’s cancer requires careful examination to determine correct and exact targets for use. These targeted molecular therapies can be advantageous, as they have the potential to cause less damage to normal cells, with fewer overall side effects. One such targeted drug is the class of menin inhibitors for *KMT2A*-rearranged AML.

### 4.1. Menin Inhibitors

The rearrangement of *KMT2A* (MLL1) occurs in up to 10% of acute leukemias and is especially common in infant leukemia, with an occurrence rate of at up to 80% [76]. Menin inhibitor treatment disrupts the interactions between menin and *KMT2A.* The transcriptional regulator *KMT2A* contains binding domains for menin, where it forms the menin-KMT2A-LEDGF complex by menin binding to menin binding domains (MBDs). This complex links *KMT2A* to chromatin, and the menin-KMT2A complex plays a crucial role in the regulation of HOX genes of hematopoiesis, specifically the leukemogenic HOXA9 and MEIS1 co-factor in myeloid progenitor stem cells [77]. In *KMT2A* rearrangement (*KMT2Ar*), this AML-sustaining fusion protein occurs and causes the overexpression of HOXA9/MEIS1 [78]. *KMT2Ar* AML carries a poor prognosis and is associated with a higher relapse rate [76]. As a cofactor, menin is necessary for *KMT2A* to bind to HOX gene promoters, and this fact has pushed the development of KMT2A-menin interaction-targeting small molecule inhibitors. Notably, menin inhibitors may be beneficial for the treatment of overexpression of other HOX genes in other subtypes of leukemia [79,80].

Menin inhibitors currently undergoing clinical trials include SNDX-5613 and KO-539, among others. The SNDX-5613 phase 1 study enrolled both adult and pediatric patients with relapsed or refractory acute leukemia, with a focus on patients with *KMT2Ar* or *NPM1*-mutated AML [24]. SNDX-5613 has been generally well tolerated thus far, with the most common side effects including QTc prolongation, nausea, and diarrhea [76]. For patients with *KMT2Ar* leukemia, the overall response rate was 23 of 38 patients (61%), with complete remission in 9 of 38 patients (24%). For patients with the *NPM1* mutation, the overall response rate was 5 of 13 (38%), and complete remission was achieved in 3 of 13 patients (23%). KO-539 is another small molecule inhibitor of the binding site of menin-KMT2A. MI-3454 is a structural analog of the KO-539 clinical compound. MI-3454 was found to have a selectivity greater than 100-fold for *KMT2Ar* cell lines compared to non-*KMT2Ar*. Unlike SNDX-5613, there was no evidence of QT prolongation or other electrocardiographic abnormalities with KO-539 [76]. The common side effects include nausea, diarrhea, and rash. In early reports, two patients out of eight (25%) achieved complete remission with KO-539 [81]. These trials continue to accumulate data, and whether some adverse events are class effects or related to individual compounds remains to be seen.

### 4.2. Tumor Suppressor Targets

The *TP53* tumor suppressor gene is frequently inactivated in cancers by loss-of-function or missense DNA binding domain mutations, seen in almost 50% of tumors [82,83], in up to 15% of overall AML cases, and in 25% of elderly cases [84]. These mutations can include loss of function or lead to protein unfolding and loss of DNA binding via substitutions of amino acid residues important for the structure of the core domain, specifically the ones making direct contact with the DNA [21]. The p53 protein additionally regulates many transcriptional targets involved in anti-angiogenesis, senescence, cell-cycle arrest, DNA repair, and apoptosis; hence, the inactivation of *TP53* allows for the enhanced proliferation of cancerous cells [21]. Targeting of mutant p53 to restore function could be a promising method for new therapeutics. APR-246 is a mutant p53 re-activating compound. It gets converted to methylene quinuclidinone (MQ), a reactive electrophile that covalently binds to the p53 core domain, and presumably, cysteine 277 is an ideal target for MQ to bind on p53 [85]. APR-246 also increases oxidative stress and induces ferroptosis as part of its anti-cancer mechanism of action [84].

APR-246 has been tested in two clinical trials, in combination with azacitidine, showing synergistic cytotoxicity in the AML cell lines with *TP53* mutations, as well as in vivo models [86]. Of the 100 patients across the two clinical trials, there was an overall response rate of 69% and a complete remission rate of 43% [85]. The phase 3 trial comparing APR-246 with azacitidine compared to azacitidine alone has been completed, but is still under evaluation, and has not been published as of this date.

Another class of targets are MDM2 (murine double minute 2 or human homolog HDM2) inhibitors that act as a physiologic antagonist of p53, which activates p53 and induces apoptosis or cell cycle arrest in wild-type p53-expressing AML. However, this could not be employed for the mutant p53. Despite this, these inhibitors have potential as p53 enhancers to initiate a tumor suppression effect in AML patients [87,88].

### 4.3. Apoptotic Inhibitors

#### MCL-1 Inhibitors

MCL-1, as a member of the BCL-2 family of proteins, is a pro-survival regulator of apoptosis. Proteins in this family bind to pro-apoptotic BH-3-only activators, bestowing MCL-1 with a role in cell death avoidance. Hematological malignancies were shown to be dependent, not only on MCL-1, but on BCL-2 as well [89]. Distinguishing it from other BCL-2 family members, MCL-1 has a very short half-life. MCL-1 inhibits mitochondrial outer membrane permeabilization, as well as the release of cytochrome C. Overexpression of MCL-1 is associated with drug resistance and a poor prognosis, and its central role in the regulation of the mitochondrial apoptotic pathway makes it a promising therapeutic target [90]. As overexpression also protects cancerous cells from apoptosis, this decreases the sensitivity to many common drugs used for treatment [91].

Development of MCL-1 inhibitors can be approached indirectly; for example, CDK inhibitors lead to decreased transcription of *MCL-1*, while mTOR inhibitors block MCL-1 translation [90]. Multiple MCL-1 inhibitors have entered the clinical trial phase, including but not limited to AZD5991 and S64315. The macrocyclic molecule AZD5991 is selective for MCL-1 and acts in a mitochondria-dependent manner, highly specific at a cellular level for MCL-1. AZD5991 binds to MCL-1 directly at the ligand-binding pocket, inducing cell death and reducing MCL-1 levels [92]. S63845 is another selective molecule that inhibits MCL-1. It binds with high specificity to the BH3-binding groove of MCL-1, where it activates the BAX/BAK-dependent apoptotic pathway [92].

### 4.4. XPO1 Inhibitors

Eltanexor (KPT-8602) is a second-generation inhibitor of XPO1-mediated nuclear export. This compound covalently binds to cysteine 528 in *XPO1*’s cargo-binding groove, blocking the interaction of *XPO1* with any cargo, therefore inhibiting nuclear export [93]. Compared to the first-generation XPO1 inhibitor selinexor, eltanexor has a much lower penetration across the blood-brain barrier. Due to its inability to enter the CNS, eltanexor results in fewer CNS-mediated side effects [94]. A recent clinical trial of eltanexor in 20 patients showed a 53.3% overall response rate, and a complete remission rate of 46.7% [95]. Unlike some of the targeted therapies described above, which are targeted at certain gene mutations, it is unclear which AML subtypes might show the best response to this class of drugs. Recently, in other clinical trial studies, it has been shown that in myelodysplastic syndrome (MDS) patients with *SF3B1* responded better to *XPO1* inhibition [95]. Moreover, in a recently concluded study, *XPO1* inhibition via selinexor has shown promising results in hematological malignancies in in vitro and in vivo studies [96]. Hence, these studies have confirmed that XPO1 inhibitors have a potential use as therapies for providing better remedies for AML patients.

### 4.5. Immune Checkpoint Inhibitors

In recent years, immune checkpoint inhibitors have shown promising results in various cancers, and they are becoming a strong option in the targeting of cancer cells. For example, programmed cell death protein 1 (PD-1) and its ligands PD-L1 and PD-L2 are revolutionizing cancer treatment in lung cancer, melanoma, Hodgkin’s lymphoma, and several other forms of cancer [97]. Another immune blockade T-lymphocyte-associated protein 4 (CTLA-4) antibody was used in hematologic malignancies, showing promising results [98,99]. There was a significant increase in the survival rates using immune checkpoint inhibitor treatment, which provided justification for exploring clinical trials in order to bring better cures for AML patients. There is another potential option to use these immune checkpoint inhibitors in combination with other classes of inhibitors to tackle the complexity of this cancer. So far, only a single clinical study has been published regarding the use of a checkpoint inhibitor as a monotherapy for AML patients. This study included eight AML patients, along with another ten patients with different hematologic malignancies that were treated with the anti-PD-1 antibody pidilizumab within a phase I study. The antibody was safe and well tolerated, except for one AML patient, who showed a minimal response manifested by a decrease in peripheral blasts from 50 to 5% [5,100]. A phase I study using the CTLA-4 antibody ipilimumab in 12 patients with AML has also been completed; however, it remains unpublished (NCT00039091) so far. Furthermore, another phase I study included 54 patients with refractory AML and other hematologic cancers (NCT01757639) and this also remains unpublished. There are three phase II studies (NCT02275533, NCT02532231, NCT02708641) currently underway studying PD-1 inhibition with either nivolumab or pembrolizumab as a monotherapy, with the goal of preventing relapse in remission. A full review of immunotherapy in AML is beyond the scope of this review, which focuses on targeted therapies, but the potential to combine targeted therapies and immunotherapies makes these relevant to mention.

### 4.6. Combinatorial Therapies in Development

To improve upon single agent therapies, a combinatorial approach seeks an advantage to strategically treat AML patients by targeting parallel pathways or providing synergistic tumor death [101]. The option of combining XPO1 inhibitors with apoptotic inhibitors is one of the options attempted in clinical trials to induce complete/partial remissions in six of 14 patients with refractory acute myeloid leukemia who had received a median of three prior therapies (ClinicalTrials.gov: NCT02530476) [102]. Likewise, *FLT3* inhibitors in combination with selinexor or eltanexor could be tried to address the complexity of this disease as tested in multiple myeloma [103]. The options with MDM2, either with BCL-2 or XPO1 inhibitors, could be a strong combination treatment to bring better remission rates, as seen by Nguyen et al. in regards to multiple myeloma.

Recently, other combination therapies have been tried, including pro-apoptotic inhibitors and hypomethylating agents. In a recent study that included randomly assigned and previously untreated patients (n = 431) with confirmed AML, a combinatorial therapy approach of venetoclax and azacitidine was employed [104]. The VIALE-A clinical trial (NCT02993523) was opened to treat older age (>75 years) patients who were not eligible for standard induction therapy due to their co-existing conditions. The combinatorial approach of venetoclax and azacitidine had shown promising effect in these patients with a longer overall survival of 14.7 months, as compared with 9.6 months with azacytidine alone (hazard ratio for death, 0.66; *p* < 0.001). Additionally, the incidence of remission rose among patients who received this combination therapy when compared to azacitidine alone. Hence, based on a thorough understanding of genomic heterogeneity, the use of different permutations in combination with specific inhibitors could be promising in order to address the complexity of this disease. Another advantage of developing combinatorial therapy is to act against the different mutations or against leukemic clone variations seen within the same patient.

## 5. Translational Perspective

The drug-like compounds, which have shown to be effective in other cancers, introduce a translational approach to the treatment of AML. Table 1 shows some of the selective and non-targeted drugs that were previously used for the treatment of other cancers, and after a thorough understanding of their mechanism of action, they were found to be effective in treating AML. They could be used in combination with a standard line of treatment or with monotherapies to reduce the toxicity of the standard treatment. Drug-like compounds and other different therapies, such as immune checkpoint inhibitors or CAR T cells, have been granted for clinical trials with AML patients in hopes of improving the disease burden and to significantly improve overall survival rates.

### Selective vs. Non-Selective Drugs

Drugs that are known to be selective and non-selective inhibitors (Table 1) have been used in the treatment of AML. As the name implies, selective inhibitors act on specific targets and limit off-target effects, whereas non-selective inhibitors reflect broad-spectrum targeting on different molecules, with magnified toxic effects (Figure 4). The selective inhibitors, as discussed in the combinatorial therapies section, have the potential advantage to utilize therapies in combination to target two or more specific targets. Therefore, they possess the capacity to be used as a therapy for personalized or precision medicine, with the pinpointed scope to treat AML patients with specific mutations through their targeted mechanism of action (MOA). Non-selective drugs with broad-spectrum targeting have the potential to treat patients with multiple driver mutations, and they have the plausible capacity to reduce resistance. However, the major concern with these non-specific drugs is the added toxicity to patients.

Since these inhibitors may have non-specific effects, they can be more difficult to use in combination therapies and can pose a great disadvantage in treating older patients. These drawbacks of non-selective drugs have led to the exploration of new therapeutic drugs that could be more reliable and exhibit fewer side effects, making them applicable to the specific morbidities of AML patients.

## 6. Conclusions

AML is a disease of a highly complex nature with a varied genomic landscape with a number of different mutations, and this complexity has presented a challenge to drug development for nearly five decades. However, the recent development of targeted therapies seeks to resolve this complexity. The main role of targeted therapy is to target the specific abnormality with maximum efficacy. The improved overall survival (OS) rate seen in patients with some of these agents is evidence of positive results that can grant hope to patients, scientists, and physicians. Despite the fact that not all targeted agents were discussed within the contents of this review, we have covered numerous promising agents of interest.

## 7. Future Research Directions

Overall, these targeted therapies show promising potential for AML patients. The broader and deeper molecular understanding of this disease has paved the way to address the core problems of treatment. However, there are challenges in regards to designing a proper scientific and clinical trial approach to attain the most accurate efficacy of these targeted drugs to deliver fuller benefits to patients. The phenotype of AML disease has been developed as a consequence of complex genetic and biological pathway changes, hence, addressing its complex nature would not be possible with one specific target, whereas a combinatorial approach could potentially include the various facets of this disease. The continuous and determined focus on understanding the underpinnings of molecular genetics and epigenetics, as well as the persistent surveillance of clonal evolution before and after the treatment of these targeted therapies, could potentially introduce novel changes to the treatment strategies, offering the maximum beneficial outcomes to patients of all ages.

## Figures and Tables

**Figure 1 biomedicines-11-00641-f001:**
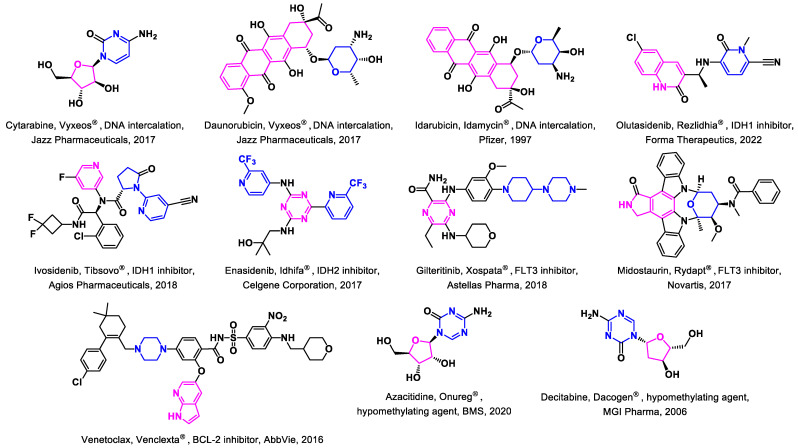
**FDA-approved drugs for the treatment of acute myeloid leukemia (AML).** The structure of standard chemotherapies—cytarabine, anthracyclines such as daunorubicin or idarubicin, and the small molecule FDA-approved drugs—included IDH inhibitors, FLT3 inhibitors, BCL-2 inhibitors, hypomethylating agents, and CPX-351. Abbreviations: DNA: deoxyribonucleic acid; IDH1/2: isocitrate dehydrogenase ½; FLT3: FMS like tyrosine kinase 3; BCL-2: B-cell lymphoma 2; BMS: Bristol Myers Squibb.

**Figure 2 biomedicines-11-00641-f002:**
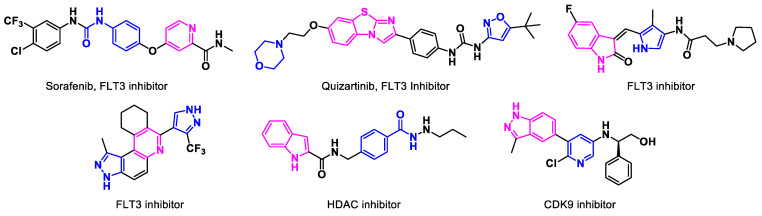
**Non-approved small molecule inhibitors for acute myeloid leukemia (AML).** In addition to the FDA-approved drugs, some other FLT3 inhibitors are currently in clinical trials. Sorafenib is a multi-targeted tyrosine kinase inhibitor, and quizartinib is a specific FLT3 inhibitor. These have not been approved by the FDA for use in AML. Abbreviations: FLT3: FMS-like tyrosine kinase 3; HDAC: histone deacetylase; CDK9: cyclin dependent kinase 9.

**Figure 3 biomedicines-11-00641-f003:**
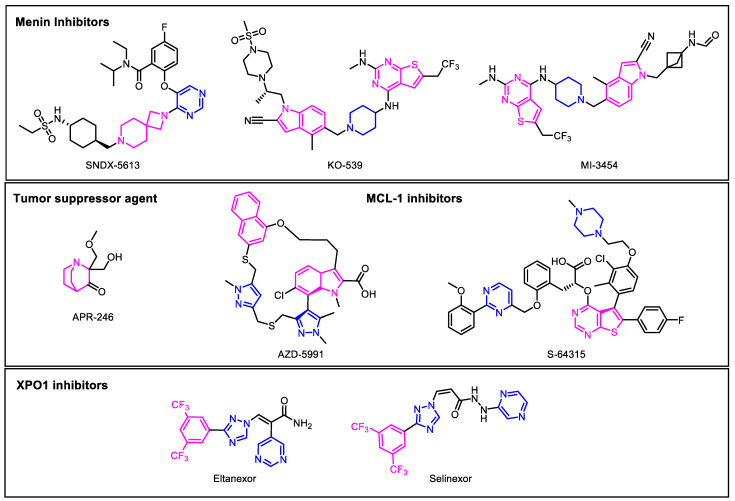
**Novel targeted therapies in development for acute myeloid leukemia (AML).** The figure shows the novel targeted therapies discussed in this review that are in development for treating AML. These include menin inhibitors, tumor suppressor agents, MCL-1 inhibitors, and XPO1 inhibitors that have recently been used for other hematologic malignancies, but have not been approved for AML. Abbreviations: MCL-1: myeloid cell leukemia 1; XPO1: exportin-1.

**Figure 4 biomedicines-11-00641-f004:**
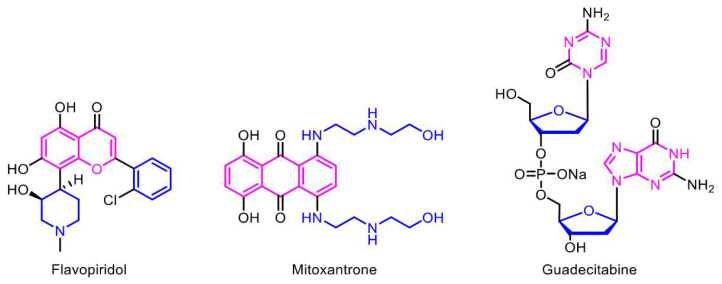
**Selective and non-selective drugs for acute myeloid leukemia (AML).** Different classes of drugs have been used for the treatment AML, which can be further classified as selective (e.g., flavopiridol) and non-selective drugs (e.g., mitoxantrone and guadecitabine), based on their targets and their mechanisms of action (MOA).

**Table 1 biomedicines-11-00641-t001:** **Target and mechanism of action (MOA) of some of the selective and non-selective drugs used in the treatment of acute myeloid leukemia (AML)**.

Selective	Target	MOA	Non-Selective	Target	MOA
Flavopiridol	CDK Inhibitor	Cell-cycle arrest and apoptosis	Daunorubicin	Anthracycline	Cytotoxic
CD33-Targeted ADCs	CD33 Target	Targeted delivery of toxic drug	Idarubicin	Anthracycline	Cytotoxic
Eltanexor	XPO1 Inhibitor	XPO1 inhibition	Mitoxantrone	Anthracycline	Topoisomerase inhibitor
Venetoclax	BCL-2 Inhibitor	Anti-apoptotic Protein inhibition	Cytarabine (CPX351)	Pyrimidine analog	DNA polymerase inhibition
Sorafenib	FLT3 Inhibitor	FLT3-ITD inhibition	Guadecitabine	Hypomethylation	DNA Methyltransferase inhibition

Abbreviations: ADC: antibody-drug conjugate; CDK: cyclin dependent kinase; XPO1: exportin-1; BCL-2: B-cell lymphoma 2; FLT3: FMS-like tyrosine kinase 3; ITD: internal tandem duplication; MOA: mechanism of action.

## Data Availability

No new data were generated.

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
