# Peer review of "Targeted Therapy Development in Acute Myeloid Leukemia"

_biomedicines, 2023, doi:10.3390/biomedicines11020641_

Round 1

Reviewer 1 Report

This paper from Totiger et al provides an extensive review of both newly approved drugs for AML patients and drugs in the horizon for new approvals. The review is rather comprehensive and unbiased, which is what a review article should attempt. However, some areas of this manuscript would benefit from further improvement.

First, and mainly most important. In the development of approved drugs, authors may not simply recapitulate the approval of the drugs and their indications, but should also offer some context to clarify their optimal use and contraindications. This would be very useful for the interested readers as clear directions are lacking.

Second, in a review, flow is as important as other aspects. The manuscript is difficult to read in some parts, so maybe some minor style checks could apply to increase readability.

Reviewer 2 Report

The review “Targeted therapy development in acute myeloid leukemia” provides a state-of-the-art overview of the currently approved and under development drugs for AML. The manuscript is mainly focused on small molecules such as IDH inhibitors, tumor suppressor agents or XPO1 inhibitors, describing their structures, MOA and clinical trials.

Although several selective and non-selective drugs are discussed, some aspects need to be deepened and the data presentation must be improved. Hence, the manuscript is suitable for publication after the following major revisions:

- Lines 33-34: the sentence is incomplete.

- At the end of lines 80-84, several citations are missing. The only one reported is too old. Add more recent studies about the “development of new small molecule drugs” in AML: Drug Dev Res. 2022 Sep;83(6):1331-1341. doi: 10.1002/ddr.21962. Epub 2022 Jun 24. PMID: 35749723; J. Med. Chem. 2020, 63, 21, 12403–12428 https://doi.org/10.1021/acs.jmedchem.0c00696. The appropriate reference is also missing at the end of lines 248 and 272.

- In analogy with the previous paragraphs, 3.1.3 should be focused on BCL2-inhibitors and a paragraph about hypomethylating agents should be added right after.

- In the figures, a different number is associated to each compound, hence they must also be reported in the text (for ex. SNDX-5613 18 or KO-539 19). Otherwise, remove them.

- Paragraphs 4.2 and 4.2.1 must be merged.

- When describing clinical trials, authors should always add the NCT number from Clinicalgov. to better identify the study.

- In the current state, paragraph 4.5 lacks relevant data. If immunotherapies need to be mentioned, add few significant results in AML.

Round 2

Reviewer 2 Report

The authors made the required corrections. The article is now suitable for publication.